# Chemical Activation of Banana Peel Waste-Derived Biochar Using KOH and Urea for CO_2_ Capture

**DOI:** 10.3390/ma17040872

**Published:** 2024-02-14

**Authors:** Joanna Sreńscek-Nazzal, Adrianna Kamińska, Jarosław Serafin, Beata Michalkiewicz

**Affiliations:** 1Department of Catalytic and Sorbent Materials Engineering, Faculty of Chemical Technology and Engineering, West Pomeranian University of Technology in Szczecin, Piastów Ave. 42, 71-065 Szczecin, Poland; joanna.srenscek@zut.edu.pl (J.S.-N.); kaminska.adrianna@zut.edu.pl (A.K.); 2Department of Inorganic and Organic Chemistry, University of Barcelona, Martí i Franquès, 1-11, 08028 Barcelona, Spain; jaroslaw.serafin@qi.ub.es

**Keywords:** banana peels, urea, activated carbon, CO_2_ adsorption

## Abstract

This article describes the synthesis and characterization of porous carbon derived from waste banana peels by chemical activation with KOH or by activation KOH and urea modification. The as-synthesized samples were carefully characterized by various techniques. The prepared carbonaceous materials possess highly developed micropore and mesopore structures and high specific surface area (up to 2795 cm^2^/g for materials synthetized with KOH and 2718 cm^2^/g for activated carbons prepared with KOH and urea). A series of KOH-activated samples showed CO_2_ adsorption at 1 bar to 5.75 mmol/g at 0 °C and 3.74 mmol/g at 25 °C. The incorporation of nitrogen into the carbon sorbent structure increased the carbon uptake capacity of the resulting materials at 1 bar to 6.28 mmol/g and to 3.86 mmol/g at 0 °C and 25 °C, respectively. It was demonstrated that treatment with urea leads to a significant increase in nitrogen content and, consequently, CO_2_ adsorption, except for the material carbonized at 900 °C. At such a high temperature, almost complete decomposition of urea occurs. The results presented in this work could be used in the future for utilization of biomass such as banana peels as a low-cost adsorbent for CO_2_ capture, which could have a positive impact on the environment and human health protection.

## 1. Introduction

In recent years, there has been a significant increase in carbon dioxide emissions into the atmosphere. It is widely believed that carbon dioxide (CO_2_) is the most dangerous anthropogenic greenhouse gas. Emissions of this gas are related to the rapid development of industrialization and technology that has taken place over the past 20 years [1]. This rapid industrial development has contributed to the overconsumption of carbon-rich compounds, such as fossil fuels, to meet energy needs, among other things. Therefore, research is being conducted to minimize CO_2_ emissions. In this regard carbon-based systems are being developed to capture carbon dioxide [2].

Activated carbons (ACs) are the most promising adsorbents proposed for CO_2_ capture due to their highly developed specific surface area and extensive pore system [3]. In addition, carbonaceous materials can be obtained from most elemental carbon-rich materials by thermal decomposition [4]. Depending on the conditions of the activation and carbonization process, as well as the type of raw material and activating agent used, the resulting carbon adsorbents can have different properties. Commonly used raw materials for industrial-scale production of activated carbon are wood, peat, bituminous coal and lignite [5]. In recent years, raw materials of plant origin (biomass) have been gaining importance as precursors for the synthesis of activated carbon. Banana peels are among the agricultural biomass waste that has potential to be used as a raw material to produce carbonaceous materials [6]. Banana peel waste is part of the solid waste generated during the processing of this fruit and is mainly produced in homes, restaurants, and food outlets. Unfortunately, this waste does not provide any economic advantages and, instead, is a pollution source, leading to environmental issues, which makes this biomass troublesome in terms of disposal and storage [7]. The application of banana peels as a precursor to production activated carbons allows the use of raw materials that have not been used before and represents a new way of utilizing these wastes. 

The characteristic feature of activated carbon that makes it so widely used in many chemical processes is the ability to modify the surface of the carbon to obtain, among other things, a material with good adsorption properties. Not surprisingly, various modifications of activated carbons are constantly being introduced to alter their acid–base, catalytic and textural properties, which largely determine their suitability for adsorption processes [8]. The advantages of carbon-based sorbents, in addition to their high concentration of micropores and large surface area, include the fact that the surface chemistry of activated carbon can be readily modified by the incorporation of various functional groups, depending on the specific requirements of the application. Additional properties, including pore size, volume, surface area and distribution, can be easily adjusted by manipulating the activation conditions and activating agents. Due to the increased availability of binding sites resulting from micropores in activated carbon, the process of CO_2_ adsorption on microporous materials is facilitated, which is another advantage of microporous adsorbents worth highlighting [1].

In order to obtain activated carbons with new and broader properties, in addition to appropriately selected chemical activation parameters, atoms of various elements are introduced into the carbon matrix. Activated carbons doped with heteroatoms have gained considerable attention due to their functionality and improvement of their remarkable properties. Some key heteroatoms such as nitrogen (N), sulfur (S) and metal oxides (MgO, CuO, etc.) are generally used to develop excellent CO_2_ adsorbents [9]. 

The physicochemical properties of nitrogen-modified activated carbons make the development of new and efficient technologies to produce such materials a major focus of functional carbon materials research at this time. From the results of studies described in the literature, it appears that the ability of activated carbons to adsorb carbon dioxide, based on the physical adsorption process, can be enhanced by incorporating nitrogen functional groups into their structure [10,11].

In general, modifications of activated carbons to enrich these materials with nitrogen can be divided into two main methods: (1) impregnation of activated carbons with nitrogen-containing compounds and (2) thermal treatment of the carbon precursor in the presence of a nitrogen-containing agent [12]. The method of impregnating activated carbons with nitrogen compounds has been widely reported in the literature. This method involves some problems related to the blocking of most of the pores present in structures of activated carbons, causing a drastic deterioration of their textural parameters [13]. The second way to synthesize nitrogen-modified carbons is to subject the precursor to thermal decomposition in the presence of a so-called reagent-N, a compound containing nitrogen atoms in its structure. The most common reagents used for this purpose are ammonia, urea, melamine and nitric oxide. To a lesser extent, pyrrole, hydrogen cyanide, formamide, hydroxylamine, hydrazine, carbazole and acridine are also used [14]. Of note is the use of urea to enrich activated carbon with nitrogen. This type of modification can be accomplished by impregnating the carbon precursor with a urea solution or by mechanically mixing the reactants in solid form. The reaction with urea allows the introduction of significant amounts of nitrogen groups on the surface of activated carbons [15]. Depending on the modification, the nitrogen content of the resulting materials is typically in the range of 4 to 9 wt.%. Activated carbons obtained by modifying the carbon precursor with urea prior to the activation process are characterized by a much more strongly developed specific surface area and a larger pore volume than analogous unmodified materials [16,17].

In this study, we report a facile strategy for synthesis of activated carbons and nitrogen-doped activated carbon by using banana peels as a carbon source, KOH as the chemical agent and urea as an additional nitrogen source through one-staged chemical activation with carbonization. It was demonstrated that treatment with urea leads to a significant increase in nitrogen content and, consequently, CO_2_ adsorption, except for the material carbonized at 900 °C. At such a high temperature, almost complete decomposition of urea occurs. Considering that the precursor was waste banana peels obtained as waste from a local fresh fruit juice production facility, a low-cost carbon source was applied and a value-added product was obtained. In addition, the use of biomass, which is difficult to dispose of, and its conversion into functional carbonaceous materials that can be used as CO_2_ sorbents, makes this waste management method environmentally friendly and in line with the so-called ‘green chemistry’ trends.

The novelty of the study lies in the description of banana peel modification using urea, which has not been previously documented. This modification resulted in increased CO_2_ adsorption. It is worth emphasizing that urea modification, activation using KOH and carbonization occurred simultaneously. CO_2_ adsorption was compared for materials produced at five different temperatures, with urea-modified and unmodified materials prepared for each temperature. This approach provided valuable insights into the nitrogen introduction capabilities using urea. Other authors typically compare urea-modified materials with a single reference material that does not contain urea.

## 2. Materials and Methods

### 2.1. Chemicals and Materials

The raw material used in the work was waste banana peels obtained from a local fresh fruit juice production point (Szczecin, Poland). The manufacturer’s stickers were removed from the banana peels, then the biomass was cut into smaller pieces and air-dried at ambient temperature for 48 h. In next step, the raw material was transferred to a laboratory dryer (Alpina, Konin, Poland) and dried at 40 °C for 24 h. The dried biomass was ground into powder in a laboratory grinder (Chemland, Stargard, Poland).

In this study, the following reagents were used: HCl (Sigma-Aldrich, Burlington, MA, USA), aqueous saturated KOH solution (Sigma-Aldrich, Burlington, MA, USA) and urea (Sigma-Aldrich, Burlington, MA, USA).

### 2.2. Synthesis of Activated Carbons from Banana Peels

Banana peels were impregnated with saturated KOH solution (the ratio of biomass to activator was 1:1) or with saturated KOH solution in the presence of urea (weight ratio 1:1) and thoroughly mixed. Impregnation of the biomass took place at ambient temperature for 3 h. After this time, the materials were placed in a laboratory dryer at 200 °C for 19 h.

The activated carbons obtained by this method were named B_KOH_x, where x corresponds to the temperature at which the carbonization process was carried out. Similarly, activated carbons obtained in the above-described method using urea were named B_KOH_urea_x, (x—carbonization temperature).

After removal from the laboratory dryer, the impregnated materials were ground, transferred to a quartz boat and placed in a tube furnace (Czylok, Jastrzębie-Zdrój, Poland).

The thermal decomposition (simultaneous carbonization and activation process) of biomass in the furnace was carried out in the temperature range of 700–900 °C and in an inert gas atmosphere (nitrogen, 18 L/h). The dwell time of the samples at the set carbonization temperature was 1 h. After the carbonization and activation process, the materials were ground again and washed with distilled water until the pH of the filtrate was close to neutral. Then, 100 mL of HCl was added to the synthesized materials and boiled. After cooling to room temperature, the samples were again washed with distilled water. The synthesized activated carbons were dried in a laboratory dryer at 180 °C.

This step was the same for the materials obtained by impregnation of biomass with saturated KOH solution and by impregnation of biomass with saturated KOH solution in the presence of urea. 

The scheme of the preparation of activated carbons is shown in Figure 1.

### 2.3. Characterization the Obtained Activated Carbons 

Nitrogen sorption at −196 °C and carbon dioxide at 0 °C and 25 °C was used to characterize textural parameters. Gas adsorption measurements were carried out on a QUADRASORB evo™ volumetric apparatus (Anton Paar, St Albans, UK; previously Quantachrome Instruments, Boynton Beach, USA, 2014). Before adsorption measurements were taken, the samples were degassed for 19 h under vacuum at 250 °C. Specific surface area was calculated from the Brunauer–Emmett–Teller (S_BET_) equation. The total pore volume was determined from the maximum adsorption of nitrogen vapor at a relative pressure of ~0.99. The density functional theory (DFT) method was used to determine the volume of micropores. 

The X-ray diffraction (XRD) analysis of the prepared activated carbons was recorded by an X-ray diffractometer (X’Pert–PRO, Panalytical, Almelo, The Netherlands, 2012) using Cu Kα radiation (λ = 0.154 nm) as the radiation source in the 2θ range of 10–100°.

Raman spectroscopy was used to determine the carbon skeleton structure of the obtained carbon materials. The Raman analysis was recorded on an InVia Raman microscope (Renishaw PLC, New Mills, Wotton-under-Edge, UK, 2007) at a laser wavelength of 785 nm. The spectra obtained in the Raman shift range from 500 cm^−1^ to 2000 cm^−1^ were analyzed. After normalizing the maximum of the G peak to 1, the intensity and position of the G and D peaks in each spectrum were read and the ratio of these intensities was determined.

To visualize the surface structures of the synthesized carbonaceous materials, the SEM pictures were taken with a SU8020 Ultra-High Resolution Field Emission Scanning Electron Microscope (Hitachi Ltd., Tokyo, Japan, 2012).

The nitrogen content of carbonaceous materials was tested using a Leco CN 628 elemental analyzer (LECO Corporation, St. Joseph, MI, USA, 2011). The standard curve was determined using an EDTA standard. Approximately 0.035 g of sample was weighed for measurement.

X-ray fluorescence (XRF) and an Epsilon 3 spectrometer (Panalytical, Almelo, The Netherlands, 2011) were utilized for elemental analysis.

## 3. Results and Discussion

Based on XRF research, it was determined that dried banana peels contain 16 wt.% potassium and 2 wt.% chlorine. The nitrogen content, tested using a Leco CN 628 elemental analyzer, was found to be 1 wt.%.

Table 1 shows the percentage yield of carbonaceous materials. The yield of activated carbon decreases as the temperature of biomass carbonization increases.

Table 2 shows the textural characterization of the carbon materials. Based on the N_2_ adsorption data, the BET surface area (S_BET_) and the pore and micropore volumes were calculated for the two groups of activated carbons studied. For KOH-activated materials, the size of the specific surface area is in the range of 1268–2795 m^2^/g, and the pore volume ranges from 0.547 to 1.652 cm^3^/g. On the other hand, for materials obtained by chemical activation with KOH in the presence of urea, the S_BET_ parameter is in the range from 1653 to 2718 m^2^/g and the total pore volume ranges from 0.718 to 1.487 cm^3^/g.

As can be seen, carbonaceous materials obtained from precursors that have been modified with urea prior to the activation process show a much more strongly developed specific surface area and a larger pore volume than analogous materials obtained from precursors that have not been treated with urea [16,18].

The use of urea significantly influenced the development of the porous structure of the tested carbon materials, as shown by the data presented in Table 1. This is particularly evident in the case of carbon B_KOH_urea_750, whose specific surface area increased by more than 600 m^2^/g after the introduction of urea into the system compared to the system without urea (B_KOH_750). In addition, for samples obtained at temperatures above 800 °C, no significant differences in textural parameters were observed, regardless of the method used to prepare the carbonaceous materials.

Since we have examined the nitrogen content in dried banana peels (1.0 wt.%), we can conclude that we are dealing with two nitrogen sources. The nitrogen content in materials untreated with urea after the carbonization process was higher than in dry banana peels. Therefore, from 0.5 to 1.1% of nitrogen was incorporated into the carbon due to contact with gaseous nitrogen at high temperatures.

The nitrogen content in materials treated with nitrogen was significantly higher. The increase in these values is attributed to nitrogen from urea. In samples treated with urea, part of the nitrogen likely comes from gaseous nitrogen. However, it is not possible to determine these proportions. Estimating this based on the difference in nitrogen content for samples carbonized at the same temperature in the presence or absence of urea would be too much of a simplification.

The textural properties of the biomass-based activated carbons were determined by N_2_ sorption at −196 °C. N_2_ adsorption–desorption isotherms for carbon materials activated with KOH are shown in Figure 2a while those for carbon materials activated with urea are shown in Figure 2b. As can be seen, both materials prepared with potassium hydroxide as the activating agent and materials with urea introduced are microporous with moderate development of mesoporosity.

Isotherms obtained from N_2_ adsorption for materials activated with KOH as well as KOH-activated materials in the presence of urea in the temperature range of 700–800 °C can be classified as Type I, while materials obtained at 850 °C and 900 °C show mixed features of Type I and IV isotherms, according to the UPAC classification [19]. 

The type I isotherm is associated with microporous materials, while the type IV isotherm indicates mesoporous materials. The initial part of the isotherm is type I with significant adsorption at low relative pressures (P/Po < 0.1), which corresponds to adsorption in micropores. At medium and high relative pressures, the isotherm is classified as type IV with an H4-type hysteresis loop, indicating that these are materials containing both micro and mesopores in their structure [19,20].

N_2_ adsorption increases as the activation temperature of the carbon materials increases, indicating a gradual development of the porous structure, reaching the highest nitrogen uptake for samples activated at a temperature of 900 °C. A similar relationship has been noted and described in the literature by other researchers [21,22].

It can be concluded that the obtained activated carbons have both micropores and mesopores, as confirmed by the distribution of pores shown in Figure 3a,b. Figure 3a,b show the pore size distribution (PSD) of KOH-activated and KOH and urea-activated carbon materials determined by DFT from N_2_ adsorption at liquid nitrogen temperature. 

As shown in Figure 3, all carbon materials contained both micropores up to the 2 nm range and mesopores in their structure. All materials, regardless of their heat treatment temperature contained mesopores in the range of 2 nm to 5 nm. However, a significant increase in the volume of mesopores in the range of 2.5–13 nm is observed for materials activated with both KOH and KOH with urea at the two highest temperatures of 850 °C and 900 °C.

When the activation temperature was higher than 800 °C, the development of mesopores was more pronounced. It can be assumed that thermal degradation of the pore wall led to the expansion of micropores to mesopores [22].

The pore size distributions shown in Figure 4a and b were determined for the carbon materials using DFT based on CO_2_ adsorption at 0 °C and confirmed the presence of micropores in all samples. The course of pore distribution for KOH-activated carbons as well as for KOH-activated carbons with urea addition is similar. The samples showed three dominant diameter ranges: 0.3–0.4 nm, 0.4–0.7 nm and 0.7–1 nm.

Nitrogen, the source of which was urea, was characterized using tests with an elemental analyzer. The results of the tests on the nitrogen content of the materials obtained are shown in Table 1. For materials obtained without urea, nitrogen content varies from 1.6–2.0 wt.%. Trends were observed to decrease with increasing temperature. The highest nitrogen content (2.0 wt.%) was characterized by material synthetized at 700 °C (B_KOH_700), while for the sample prepared at 900 °C (B_KOH_900), the value was 1.6%. The nitrogen content of materials unmodified with urea is due to the fact that this element occurs naturally in banana peels (the percentage of nitrogen in this biomass noted in the literature is estimated at about 2% wt; [23]), and was not removed from the carbonaceous material during the carbonization process.

Much greater differences in nitrogen content were observed for urea-modified samples. The highest nitrogen content of 5.8 wt.% was noted for activated carbon carbonized at 700 °C (B_KOH_urea_700), while for the sample prepared at the highest temperature (B_KOH_urea_900), the value was only 2.0 wt.%. 

From the data presented in Table 1, it can be concluded that the amount of nitrogen introduced depends on the carbonization temperature; more nitrogen was introduced into activated carbons obtained at lower temperatures. Similar observations have been reported in the literature for orange peel carbons prepared with urea [24], for N-doped activated carbons from bamboo charcoal [25] and for commercial urea-modified activated carbons [26]. The decrease in nitrogen content with increasing carbonization temperature is due to the degradation of unstable thermal nitrogen groups at higher process temperatures [27].

XRD analysis was employed to characterize crystallinity of the carbon materials, and the obtained patterns are presented in Figure 5a,b.

XRD profiles show that all materials are low crystalline. The absence of sharp diffraction peaks proves that there is no crystalline phase in the materials. Comparing the data shown in Figure 5, no significant differences were found between the materials prepared with KOH and those obtained with KOH in the presence of urea. Fr all carbons, the diffraction patterns show only relatively broad diffraction peaks at ~43°, and for the B_KOH samples, a small broad diffraction peak appeared at ~24° the which correspond to the 100 and 002 diffraction planes of carbon, respectively. The peak at ~24° for the other materials disappeared (flattened out). The low intensity of the peak corresponding to the plane (1 0 0) at approx. 44° for activated carbons indicates that these materials have an amorphous, highly disordered structure, which is characteristic of carbon materials [28]. Similar diffractograms of poorly available carbon materials were reported in [29,30], in which the authors attributed them to amorphous carbons. 

Structural properties of B_KOH and B_KOH_urea were also tested with Raman spectra. Raman spectra for activated carbons synthesized at temperatures ranging from 700 °C to 900 °C are shown in Figure 6a,b.

The spectra were characterized by two broad bands: the D band located at 1300 cm^−1^, attributed to the disordered phase, and the G band located at 1600 cm^−1^, corresponding to the crystalline phase. It was found that the D band was more intense than the G band, indicating that the structure of carbon was significantly disordered and associated with the presence of numerous defects in the structure [29].

In addition, the I_G_/I_D_ ratios of all prepared activated carbons are shown in Figure 6. The ratio of the intensity of the G band to the intensity of the D band (I_G_/I_D_) indicates the degree of graphitization of carbon materials. An increase in the I_G_/I_D_ peak intensity ratio indicates a higher degree of graphitization (ordering) of the carbonaceous material.

Activated carbon samples showed similar I_G_/I_D_ ratios in the range of 0.624–0.693 for materials prepared with KOH and in the range of 0.656–0.730 for materials obtained with KOH and urea. According to the I_G_/I_D_ ratios, sample B_KOH_700 (I_G_/I_D_ = 0.693) and sample B_KOH_urea_700 (I_G_/I_D_ = 0.730) were more ordered materials than sample B_KOH_900 (I_G_/I_D_ = 0.624) and sample B_KOH_urea_900 (I_G_/I_D_ = 0.656), indicating that the degree of ordering of the materials decreases as the carbonization temperature increases. 

The morphology of the carbon materials was characterized by SEM, as shown in Figure 7.

The activated carbons prepared from banana peel biomass showed a porous structure with typical cavities. The SEM images show a disordered structure, which is typical of activated carbons. The materials contain numerous large tunnels, holes and channels [31]. A similar morphology of activated carbons produced from biomass has been reported in other studies [4,31].

The CO_2_ adsorption capacity of this series of sorbents was investigated under 1 bar at 0 °C and at 25 °C. Figure 8 shows the results of CO_2_ adsorption tests, and corresponding numerical data are provided in Table 3.

As can be seen from the data presented in Table 3 and Figure 8 for all activated carbons, regardless of the synthesis method, the CO_2_ adsorption decreased with increasing carbonization temperature. In addition, for activated carbons activated with KOH and urea, the CO_2_ adsorption was higher than for carbons activated only with KOH. Sorbents activated only with KOH possessed CO_2_ uptake ranging from 2.64 to 3.71 mmol/g at 25 °C and 5.06 to 5.75 mmol/g at 0 °C under 1 bar. In contrast, nitrogen-doped carbon materials showed CO_2_ adsorption in the range of 2.82 to 3.85 mmol/g at 25 °C and from 4.77 mmol/g to 6.26 mmol/g at 0 °C under 1 bar. Among all samples, B_KOH_urea_750 had the highest CO_2_ capture adsorption at 0 °C and 25 °C under 1 bar at 6.26 mmol/g and 3.85 mmol/g, respectively. The CO_2_ adsorption obtained on this material was significantly higher compared to material prepared under the same conditions but without urea.

Based on the results, it can be concluded that the use of urea in the synthesis of activated carbons contributes to an increase in the CO_2_ adsorption capacity of samples obtained by this method, compared to samples obtained using only KOH as a chemical activator (Figure 9). The exception is materials obtained at 900 °C. The adsorption of CO_2_ on the activated carbon not treated with urea is higher. This can be explained by the higher values of the textural parameters of this sample. It is clear that CO_2_ adsorption is affected simultaneously by several parameters: nitrogen content and values of textural parameters. The introduction of urea also affects the modification of textural parameters. For a given temperature, it causes an increase in the volume of micropores and specific surface area except at 900 °C for S_BET_. An increase in pore volume was observed only for samples obtained in the temperature range of 700–800 °C. It is not possible to analyze the effect on adsorption based solely on nitrogen content or textural parameters alone. However, the figure clearly shows that at temperatures below 900 °C, urea treatment is beneficial.

Moreover, considering that CO_2_ is an acidic gas, the inclusion of nitrogen in the carbon matrix promotes CO_2_ capture by imparting an alkaline character to the adsorbent surface. The increase in CO_2_ sorption capacity observed for N-doped materials (B_KOH_urea_x series) is associated with an increase in the surface alkalinity of activated carbons obtained by chemical activation with KOH in the presence of urea [32]. In the literature, urea is described as a source of nitrogen, the use of which in the production process of activated carbons makes it possible to obtain materials with the highest CO_2_ capture efficiency (compared to adsorbents for which other nitrogen sources were used) [32].

Many reports describing the use of activated carbon as a CO_2_ sorbent have been published in the literature. Table 4 shows high CO_2_ adsorption at 1 bar on activated carbons produced from biomass described by other authors. To summarize, these biomass-based carbonaceous materials exhibit CO_2_ uptake ranging from 4.1 mmol/g to 7.3 mmol/g at 0 °C and from 0.5 mmol/g to 5.0 at 25 °C. 

Considering the data in the Table 4, the CO_2_ adsorption values obtained in this work on activated carbon from banana peel are not the highest values in compered to data reported in the literature, but it should be noted that the values reported in this work are among the highest for biomass-derived activated carbons. 

The presence of heteroatoms, such as nitrogen, can also affect the adsorption capacity of activated carbon, and this phenomenon has also been described in the literature. Table 5 shows the CO_2_ adsorption on activated carbons obtained by activation with KOH and different N-reagents or activation with N-reagent only.

According to the data presented in the table, the activated carbon obtained by our team (by activating KOH in the presence of urea as a nitrogen source) has a CO_2_ adsorption value at a level comparable to, and in some cases greater than, the materials described in other works and obtained by the same method.

In addition, the CO_2_ adsorption values for activated carbons obtained by activation with N-reagent without activating agent in the form of KOH are significantly lower (3.4 mmol/g at 0 °C and 2.3 mmol/g at 25 °C) than the adsorption values obtained for other carbon materials (6.3 mmol/g at 0 °C and 3.9 mmol/g at 25 °C), which proves that the method chosen by our team to obtain activated carbons from biomass is the best way to obtain materials that can be applied in carbon adsorption.

## 4. Conclusions

It has been demonstrated that waste obtained from a local juice production facility, specifically banana peels, serves as a good precursor for the production of activated carbon intended for CO_2_ adsorption. It has been shown that the addition of urea enhances CO_2_ adsorption. Simultaneous treatment of banana peels with KOH and urea leads to an increased nitrogen content in the resulting product, which is activated carbon. Unquestionably, it has been demonstrated that the increase in nitrogen content correlates with an elevation in CO_2_ adsorption. On the other hand, raising the carbonization temperature causes the decomposition of nitrogen-containing groups on the surface of activated carbon. We observed that the optimal temperature for producing activated carbon with an elevated nitrogen content is 750 °C. The highest CO_2_ adsorption under 1 bar pressure was achieved with the B_KOH_urea_750 material, which reached 6.36 mmol/g at 0 °C and 3.85 mmol/g at 25 °C.

## Figures and Tables

**Figure 1 materials-17-00872-f001:**
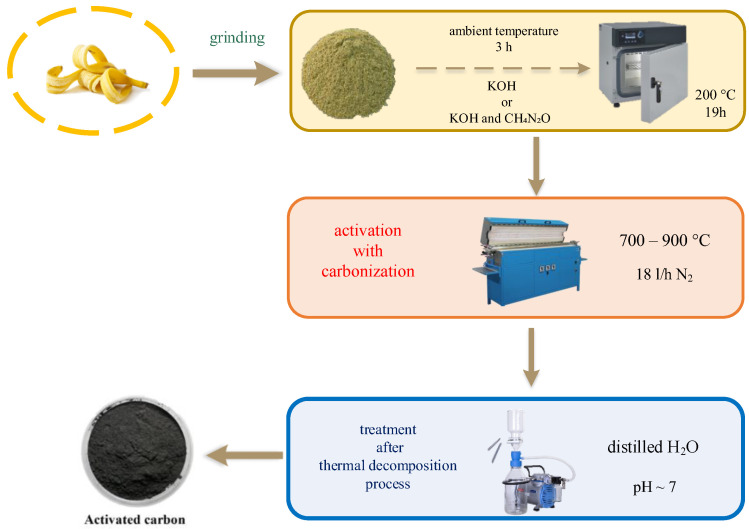
Schematic illustration for synthesis of banana-derived porous carbons.

**Figure 2 materials-17-00872-f002:**
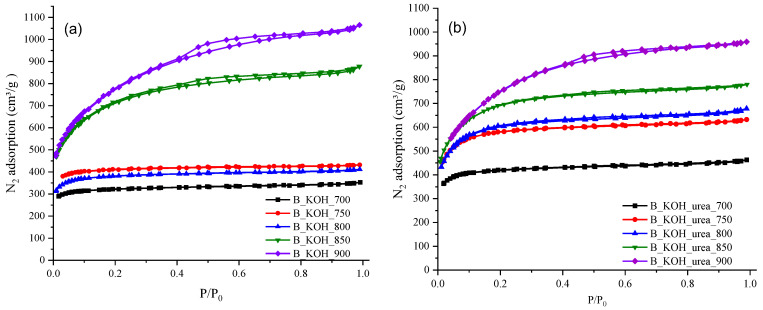
Adsorption–desorption isotherms of (**a**) samples synthesized with KOH and (**b**) for samples synthesized with KOH and urea.

**Figure 3 materials-17-00872-f003:**
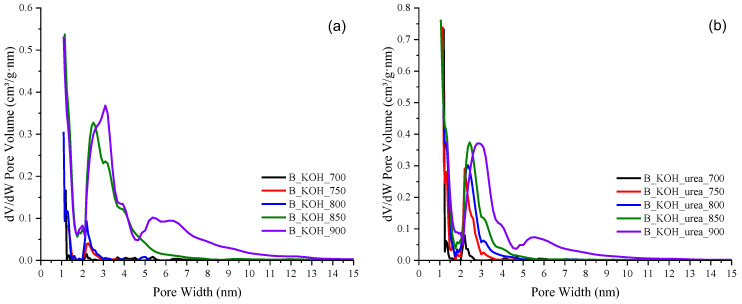
Carbon pore size distribution determined by DFT method for (**a**) samples synthesized with KOH and (**b**) samples synthesized with KOH and urea.

**Figure 4 materials-17-00872-f004:**
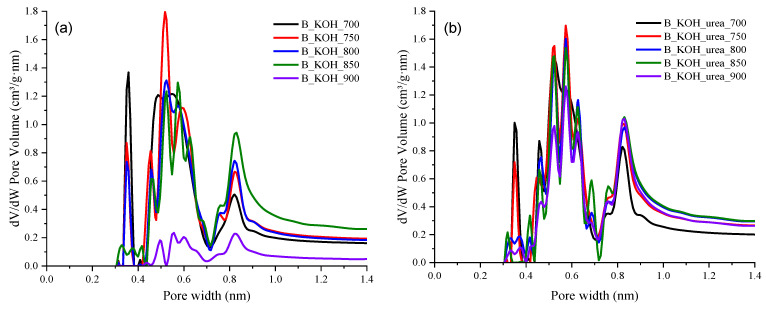
Micropore size distribution curves of activated carbons prepared (**a**) with KOH (**b**) with KOH and urea calculated on the basis of CO_2_ adsorption.

**Figure 5 materials-17-00872-f005:**
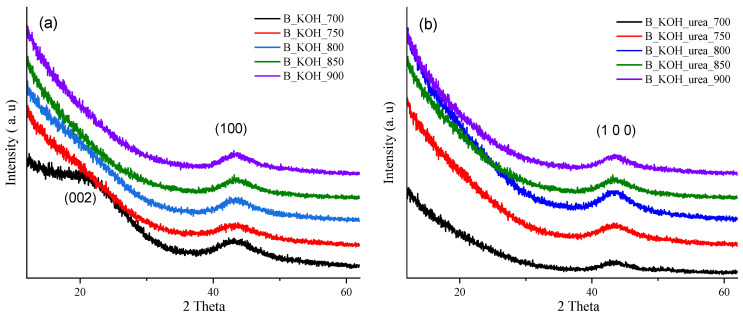
Diffraction patterns of the activated carbons synthesized from banana with (**a**) KOH and (**b**) KOH with urea and carbonized under various temperatures.

**Figure 6 materials-17-00872-f006:**
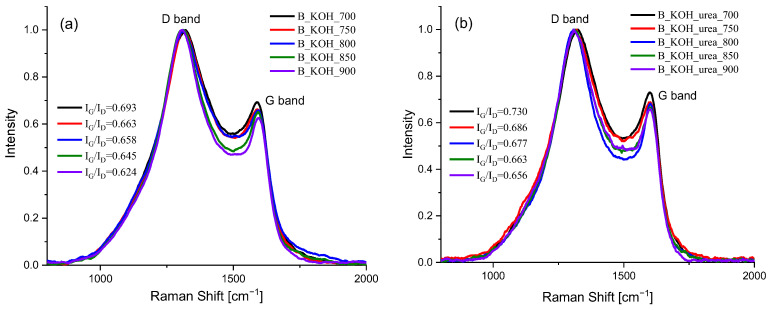
Raman spectra of activated carbon samples produced from banana peels with (**a**) KOH and (**b**) KOH und urea.

**Figure 7 materials-17-00872-f007:**
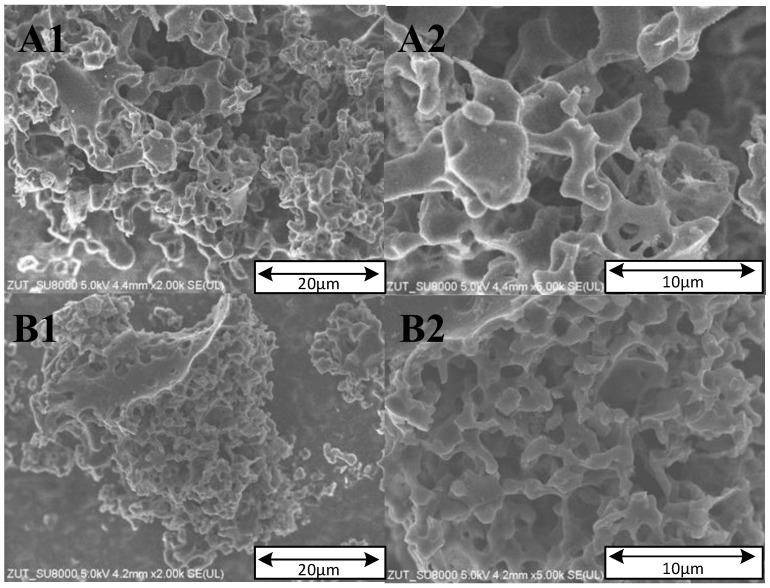
SEM micrographs of the activated carbons: (**A1**,**A2**) activated carbon from banana peels prepared with KOH; (**B1**,**B2**) activated carbon from banana peels prepared with KOH and urea.

**Figure 8 materials-17-00872-f008:**
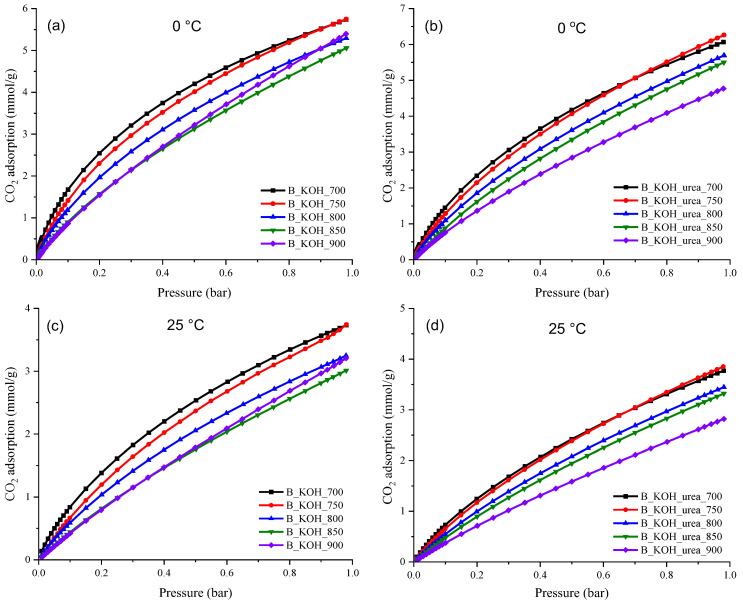
CO_2_ adsorption isotherms measured at 0 °C for (**a**) B_KOH series and (**b**) B_KOH_urea series of carbons and CO_2_ adsorption isotherms measured at 25 °C for (**c**) B_KOH series and (**d**) B_KOH_urea series.

**Figure 9 materials-17-00872-f009:**
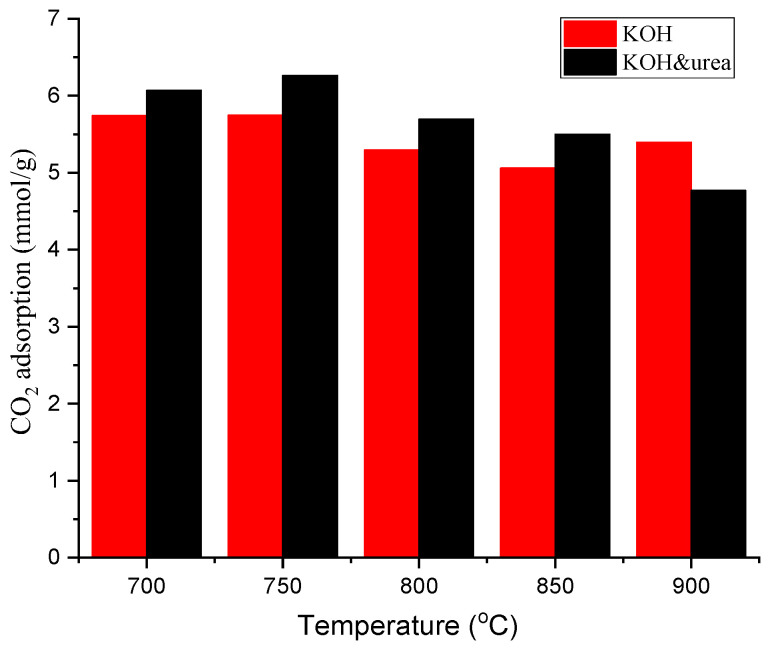
CO_2_ adsorption isotherms measured at 0 °C and 1 bar.

**Table 1 materials-17-00872-t001:** Percentage yield of carbonaceous materials.

Sample	Yield (%)
B_KOH_700	15
B_KOH_750	15
B_KOH_800	14
B_KOH_850	11
B_KOH_900	11
B_KOH_urea_700	19
B_KOH_urea_750	18
B_KOH_urea_800	15
B_KOH_urea_850	10
B_KOH_urea_900	10

**Table 2 materials-17-00872-t002:** Values of the textural parameters of carbonaceous materials.

Sample	S_BET_ (m^2^/g)	Vtot (cm^3^/g)	Vmic (cm^3^/g)	N Content (wt.%)
B_KOH_700	1268	0.547	0.464	2.1
B_KOH_750	1623	0.669	0.582	1.5
B_KOH_800	1651	0.744	0.565	1.7
B_KOH_850	2598	1.361	0.680	1.7
B_KOH_900	2795	1.652	0.608	1.6
B_KOH_urea_700	1653	0.718	0.596	5.8
B_KOH_urea_750	2228	0.981	0.726	3.6
B_KOH_urea_800	2271	1.052	0.690	2.9
B_KOH_urea_850	2580	1.209	0.731	2.5
B_KOH_urea_900	2718	1.487	0.632	2.1

**Table 3 materials-17-00872-t003:** Results of the adsorption experiments at CO_2_ pressure of 1 bar and at 0 °C and 25 °C.

CO_2_ Uptake (mmol/g)		
Sample	at 0 °C	at 25 °C
B_KOH_700	5.74	3.73
B_KOH_750	5.75	3.74
B_KOH_800	5.29	3.25
B_KOH_850	5.06	3.01
B_KOH_900	5.40	3.21
B_KOH_urea_700	6.07	3.77
B_KOH_urea_750	6.26	3.85
B_KOH_urea_800	5.70	3.45
B_KOH_urea_850	5.50	3.32
B_KOH_urea_900	4.77	2.82

**Table 4 materials-17-00872-t004:** Reported CO_2_ capture results for activated carbons prepared from biomass by chemical activation by KOH.

Activated Carbon Precursor	CO_2_ Adsorption at 0 °C [mmol/g]	CO_2_ Adsorption at 25 °C [mmol/g]	References
Vine Shoots	5.4	-	[33]
*Arundo donax*	4.1	3.2	[34]
Carrot Peels	5.6	4.2	[35]
Celtuce Leaves	6.0	4.4	[36]
Coffee Grounds	4.4	3.0	[37]
Lumpy Bracket	7.2	4.6	[38]
Peanut Shell	7.3	4.4	[39]
Pomegranate Peels	6.0	4.1	[35]
*Sargassum*	-	1.1	[40]
Palm Shells	6.3	4.4	[41]
Fern Leaves	4.5	4.1	[35]
*Enteromorpha*	-	0.5	[40]
Bee-Collected Pollen	5.6	4.2	[42]
Molasses	5.4	-	[43]
Sawdust	6.6	4.3	[44]
Starch	5.6	3.5	[44]
Cellulose	5.8	3.5	[44]
Jujun Grass	-	4.9	[45]
Wood of *Camellia japonica*	-	5.0	[45]
Banana peels	5.75	3.74	This work

**Table 5 materials-17-00872-t005:** Comparison of the CO_2_ adsorption for recent N-doped activated carbons obtained from biomass-based materials.

Activated Carbon Precursor	Activation with KOH and N-Reagent	Activation with N-Reagent	CO_2_ Adsorption at 0 °C [mmol/g]	CO_2_ Adsorption at 25 °C [mmol/g]	References
Coconut shell	KOH + Urea	-	7.00	5.00	[46]
Black locust	KOH + Ammonia	-	-	5.05	[47]
Longan shell	KOH+ Carbamide	-	5.60	4.30	[48]
Walnut shell	KOH + Urea	-	-	5.4	[49]
Chestnut tannin	-	NH_3_ activation	3.44	2.27	[50]
Corncob particles	-	NH_3_ activation	4.50	2.81	[51]
Banana peels	KOH + Urea	-	6.26	3.86	This work

## Data Availability

Data are available upon request to the corresponding author.

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
