# Peer review of "Chemical Activation of Banana Peel Waste-Derived Biochar Using KOH and Urea for CO2 Capture"

_materials, 2024, doi:10.3390/ma17040872_

Round 1

Reviewer 1 Report

Comments and Suggestions for Authors

 The manuscript “Chemical Activation of Banana Peels Waste-derived Biochar using KOH and Urea for CO2 Capture” of authors Joanna SreÅ„scek-Nazzal, Adrianna KamiÅ„ska, JarosÅ‚aw Serafin and Beata Michalkiewicz is very interesting. I think the work can be published after minor revision.

 I have some comments:

1.       The authors did not refer to the losses of ammonia or nitrogen oxides from urea during the heat treatment.

2.       It is also necessary to specify the source of nitrogen (urea or the atmosphere in the treatment oven at 700-900 degrees. The authors must add a Raman characterization of the materials after the treatment at 200 degrees.

3.       In introduction the authors written:” The prepared carbonaceous materials possess highly developed micropores structures….” and in text at raw 220:” It can be concluded that the obtained activated carbons have both micropores and mesopores, as confirmed by the distribution of pores shown in Figure 3a and Figure 3b.” The authors must describe the obtained results homogeneously.

4.       The authors compared the absorption properties with those of other types of materials. They did not highlight enough the positive differences in the absorption of the material obtained by them.

5.       In conclusions the phrases “Simultaneous treatment of banana peels with KOH and urea leads to an increased nitrogen content in the resulting product, which is activated carbon.” The authors must rewrite the conclusions.

6.       Minor English mistakes.

7.       The authors must check carefully the manuscript.

Comments on the Quality of English Language

Minor revision

Author Response

Answers in file

Reviewer 2 Report

Comments and Suggestions for Authors

The comments are here attached.

Author Response

Answers in file
